# MASKVIT: MASKED VISUAL PRE-TRAINING FOR VIDEO PREDICTION

**Agrim Gupta**[1]*, **Stephen Tian**[1], **Yunzhi Zhang**[1], **Jiajun Wu**[1], **Roberto Martín-Martín**[1,2,3],
**Li Fei-Fei**[1]
[1]Stanford University, [2]UT Austin, [3]Salesforce AI

## ABSTRACT

The ability to predict future visual observations conditioned on past observations and motor commands can enable embodied agents to plan solutions to a variety of tasks in complex environments. This work shows that we can create good video prediction models by pre-training transformers via masked visual modeling. Our approach, named MaskViT, is based on two simple design decisions. First, for memory and training efficiency, we use two types of window attention: spatial and spatiotemporal. Second, during training, we mask a *variable* percentage of tokens instead of a *fixed* mask ratio. For inference, MaskViT generates all tokens via iterative refinement where we incrementally decrease the masking ratio following a mask scheduling function. On several datasets we demonstrate that MaskViT outperforms prior works in video prediction, is parameter efficient, generates high-resolution videos ($256 \times 256$) and can be easily adapted to perform goal-conditioned video prediction. Further, we demonstrate the benefits of inference speedup (up to $512\times$) due to iterative decoding by using MaskViT for planning on a real robot. Our work suggests that we can endow embodied agents with powerful predictive models by leveraging the general framework of masked visual modeling with minimal domain knowledge.

## 1 INTRODUCTION

Evidence from neuroscience suggests that human cognitive and perceptual capabilities are supported by a predictive mechanism to anticipate future events and sensory signals (Tanji & Evarts, 1976; Wolpert et al., 1995). Such a mental model of the world can be used to simulate, evaluate, and select among different possible actions. This process is fast and accurate, even under the computational limitations of biological brains (Wu et al., 2016). Endowing robots with similar predictive capabilities would allow them to plan solutions to multiple tasks in complex and dynamic environments, e.g., via visual model-predictive control (Finn & Levine, 2017; Ebert et al., 2018).

Predicting visual observations for embodied agents is however challenging and computationally demanding: the model needs to capture the complexity and inherent stochasticity of future events while maintaining an inference speed that supports the robot's actions. Therefore, recent advances in autoregressive generative models, which leverage Transformers (Vaswani et al., 2017) for building neural architectures and learn good representations via self-supervised generative pretraining (Devlin et al., 2019), have not benefited video prediction or robotic applications. We in particular identify three technical challenges. First, memory requirements for the full attention mechanism in Transformers scale quadratically with the length of the input sequence, leading to prohibitively large costs for videos. Second, there is an inconsistency between the video prediction task and autoregressive masked visual pretraining – while the training process assumes *partial* knowledge of the ground truth future frames, at test time the model has to predict a complete sequence of future frames from *scratch*, leading to poor video prediction quality (Yan et al., 2021; Feichtenhofer et al., 2022). Third, the common autoregressive paradigm effective in other domains would be too slow for robotic applications.

To address these challenges, we present **Mask**ed **Vi**deo **T**ransformers (MaskViT): a simple, effective and scalable method for video prediction based on masked visual modeling. Since using pixels

---

*Correspondence to `agrim@stanford.edu`

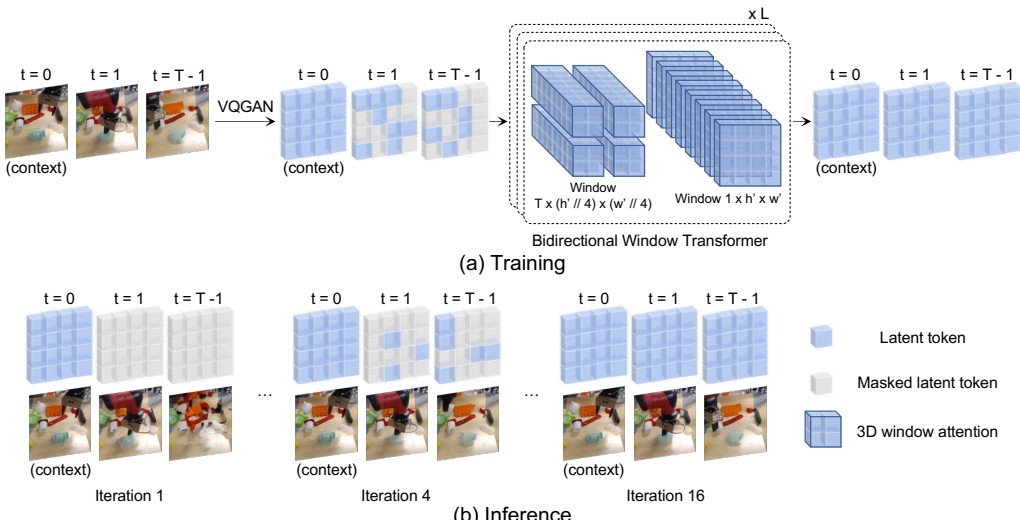

Figure 1: **MaskViT**. (a) Training: We encode the video frames into latent codes via VQ-GAN. A *variable* number of tokens in future frames are masked, and the network is trained to predict the masked tokens. A block in MaskViT consists of two layers with window-restricted attention: spatial and spatiotemporal. (b) Inference: Videos are generated via iterative refinement where we incrementally decrease the masking ratio following a mask scheduling function. Videos available at this project page.

directly as frame tokens would require an inordinate amount of memory, we use a discrete variational autoencoder (dVAE) (Van Den Oord & Vinyals, 2017; Esser et al., 2021) that compresses frames into a smaller grid of visual tokens. We opt for compression in the spatial (image) domain instead of the spatiotemporal domain (videos), as preserving the correspondence between each original and tokenized video frame allows for flexible conditioning on any subset of frames – initial (past), final (goal), and possibly equally spaced intermediate frames. However, despite operating on tokens, representing 16 frames at 256 tokens per frame still requires 4, 096 tokens, incurring prohibitive memory requirements for full attention. Hence, to further reduce memory, MaskViT is composed of alternating transformer layers with non-overlapping *window-restricted* (Vaswani et al., 2017) spatial and spatiotemporal attention.

To reduce the inconsistency between the masked pretraining and the video prediction task and to speed up inference, we take inspiration from non-autoregressive, iterative decoding methods in generative algorithms from other domains (Sohl-Dickstein et al., 2015; Ho et al., 2020; Nichol & Dhariwal, 2021; Ghazvininejad et al., 2019; Chang et al., 2022). We propose a novel iterative decoding scheme for videos based on a mask scheduling function that specifies, during inference, the number of tokens to be decoded and kept at each iteration. In contrast to autoregressive decoding, which involves predicting tokens one by one, our iterative decoding scheme is faster as the number of decoding iterations is significantly less than the number of tokens. A few initial tokens are predicted over multiple initial iterations, and then the majority of the remaining tokens can be predicted rapidly over the final few iterations. This brings us closer to the ultimate video prediction task, where only the first frame is known and all tokens for other frames must be inferred. To further close the training-test gap, during training we mask a *variable* percentage of tokens, instead of using a *fixed* masking ratio. This simulates the different masking ratios MaskViT will encounter during iterative decoding in the actual video prediction task.

Through experiments on several publicly available real-world video prediction datasets (Ebert et al., 2017; Geiger et al., 2013; Dasari et al., 2019), we demonstrate that MaskViT achieves competitive or state-of-the-art results in a variety of metrics. Moreover, MaskViT can predict considerably higher resolution videos ($256 \times 256$) than previous methods. We also show the flexibility of MaskViT by adapting it to predict goal-conditioned video frames. In addition, thanks to iterative decoding, MaskViT is up to $512\times$ faster than autoregressive methods, enabling its application for planning on a real robot (§ 4.5). These results indicate that we can endow embodied agents with powerful

predictive models by leveraging the advances in self-supervised learning in language and vision, without engineering domain-specific solutions.

## 2   RELATED WORK

**Video prediction.** The video prediction task refers to the problem of generating videos conditioned on past frames (Ranzato et al., 2014; Lotter et al., 2016), possibly with an additional natural language description (Li et al., 2018; Gupta et al., 2018; Pan et al., 2017; Wu et al., 2021b) and/or motor commands (Finn et al., 2016; Villegas et al., 2019; Wu et al., 2021a; Babaeizadeh et al., 2021). Multiple classes of generative models have been utilized to tackle this problem, such as Generative adversarial networks (GANs) (Clark et al., 2019; Tulyakov et al., 2018; Luc et al., 2020), Variational Autoencoders (VAEs) (Villegas et al., 2019; Wu et al., 2021a; Babaeizadeh et al., 2021; 2018; Denton & Fergus, 2018; Akan et al., 2021; 2022), invertible networks (Dorkenwald et al., 2021), autoregressive (Yan et al., 2021; Rakhimov et al., 2020; Nash et al., 2022) and diffusion (Ho et al., 2022; Voleti et al., 2022) models. Our work focuses on predicting future frames conditioned on past frames or motor commands and belongs to the family of two-stage methods that first encode the videos into a downsampled latent space and then use transformers to model an autoregressive prior (Yan et al., 2021; Rakhimov et al., 2020). A common drawback of these methods is the large inference time due to autoregressive generation. MaskViT overcomes this issue by using an iterative decoding scheme, which significantly reduces inference time.

**Masked autoencoders.** Masked autoencoders are a type of denoising autoencoder (Vincent et al., 2008) that learn representations by (re)generating the original input from corrupted (i.e., masked) inputs. Masked language modeling (MLM) was first proposed in BERT (Devlin et al., 2019) and has revolutionized the field of natural language processing, especially when scaled to large datasets and model sizes (Brown et al., 2020; Radford et al., 2019). MLM with fixed and low masking ratio has also been extended for multi-modal representation learning (Lu et al., 2019; Sun et al., 2019; Zellers et al., 2021). The success in NLP has also been replicated in vision by masking a fixed but high ratio of patches of pixels (He et al., 2021; Dosovitskiy et al., 2020) or masking tokens generated by a pretrained dVAE (Bao et al., 2022; Chen et al., 2020). Recently, these works have also been extended to video domains to learn good representations for action recognition (Tong et al., 2022; Feichtenhofer et al., 2022). Unlike them, we apply masked visual modeling for video prediction, and we use a *variable* masking ratio during training to reduce the difference between masked pretraining and video prediction. Another related line of work is leveraging good visual representations learnt via self supervised learning methods (Laskin et al., 2020; Nair et al., 2022; Parisi et al., 2022) including masked autoencoders (Xiao et al., 2022) for motor control.

**Non-autoregressive decoding.** A key limitation of autoregressive decoding is that each token is predicted sequentially. Hence, the decoding time is proportional to the dimensionality of the data. An appealing alternative is iterative decoding, where all tokens are predicted simultaneously and then refined for a fixed number of steps independent of the data dimensionality (Gu et al., 2017; Ghazvininejad et al., 2019; Saharia et al., 2020). Ghazvininejad et al. (2019) proposed a *mask-predict* decoding paradigm for machine translation by training a bidirectional decoder instead of a causal decoder. During inference, entire translated sequences are predicted in parallel within a constant number of steps. Chang et al. (2022) extend this framework to the image domain and use mask scheduling functions during training and decoding. We extend the framework of *mask-predict* (Ghazvininejad et al., 2019) and MaskGiT (Chang et al., 2022) for video prediction. We pre-train a bidirectional window transformer model via masked visual modeling where we mask a *variable* percentage of tokens independent of the mask scheduling function. Importantly, unlike MaskGiT, instead of keeping the most confident tokens during each decoding step, we add temperature annealed Gumbel noise to the token confidence to produce diverse outputs.

## 3   MASKVIT: MASKED VIDEO TRANSFORMER

MaskViT is the result of a two-stage training procedure (Van Den Oord & Vinyals, 2017; Razavi et al., 2019): First, we learn an encoding of the visual data that discretizes images into tokens based on a discrete variational autoencoder (dVAE). Next, we deviate from the common autoregressive training objective and pre-train a bidirectional transformer with window-restricted attention via

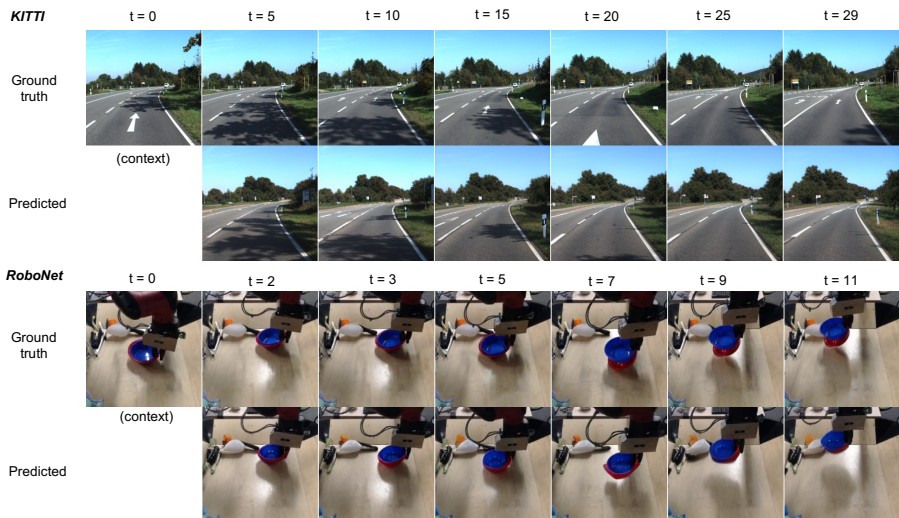

Figure 2: **High resolution video prediction.** Video prediction results on test set of KITTI and RoboNet at $256 \times 256$ resolution. See § B.2 for additional qualitative results.

*masked visual modeling* (MVM). In the following section, we describe our image tokenizer, bidirectional transformer, masked visual pre-training, and iterative decoding procedure.

## 3.1  LEARNING VISUAL TOKENS

Videos contain too many pixels to be used directly as tokens in a transformer architecture. Hence, to reduce dimensionality, we first train a VQ-VAE (Van Den Oord & Vinyals, 2017) for individual video frames so that we can represent videos as sequences of grids of discrete tokens. VQ-VAE consists of an encoder $E(x)$ that encodes an input image $x \in \mathbb{R}^{H \times W \times 3}$ into a series of latent vectors. The vectors are discretized through a nearest neighbour look up in a codebook of quantized embeddings, $\mathcal{Z} = \{z_k\}_{k=1}^K \subset \mathbb{R}^{n_z}$. A decoder $D$ is trained to predict a reconstruction of the image, $\hat{x}$, from the quantized encodings. In our work, we leverage VQ-GAN (Esser et al., 2021), which improves upon VQ-VAE by adding adversarial (Goodfellow et al., 2014) and perceptual losses (Johnson et al., 2016; Zhang et al., 2018). Each video frame is individually tokenized into a $16 \times 16$ grid of tokens, regardless of their original resolution (Fig. 1, a, left). Instead of using 3D extensions of VQ-VAE which perform spatiotemporal compression of videos (Yan et al., 2021), our per-frame compression enables us to condition on arbitrary context frames: initial, final, and possibly intermediate ones.

## 3.2  MASKED VISUAL MODELING (MVM)

Inspired by the success of masked language (Devlin et al., 2019) and image (Bao et al., 2022; He et al., 2021) modeling, and in the spirit of unifying methodologies across domains, we pre-train MaskViT via MVM for video prediction. Our pre-training task and masking strategy are straightforward: we keep the latent codes corresponding to context frames intact and mask a random number of tokens corresponding to future frames. The network is trained to predict masked latent codes conditioned on the unmasked latent codes.

Concretely, we assume access to input context frames for $T_c$ time steps, and our goal is to predict $T_p$ frames during test time. We first quantize the entire video sequence into latent codes $Z \in \mathbb{R}^{T \times h \times w}$. Let $Z_p = [z_i]_{i=1}^N$ denote the latent tokens corresponding to future video frames, where $N = T_p \times h \times w$. Unlike prior work on MVM (Bao et al., 2022; He et al., 2021) that uses a *fixed* masking ratio, we propose to use a *variable* masking ratio that reduces the gap between pre-training task and inference leading to better evaluation results (see § 3.4). Specifically, during training, for each video in a batch, we first select a masking ratio $r \in [0.5, 1)$ and then randomly select and replace $\lfloor r \cdot N \rfloor$ tokens in $Z_p$ with a `[MASK]` token. The pre-training objective is to minimize the negative log-likelihood of the visual tokens given the masked video as input:

$\mathcal{L}_{\text{MVM}} = - \mathbb{E}_{x \in \mathcal{D}} \left[ \sum_{\forall i \in N^M} \log p(z_i | Z_p^M, Z_c) \right]$, where $\mathcal{D}$ is the training dataset, $N^M$ represents randomly masked positions, and $Z_p^M$ denotes the output of applying the mask to $Z_p$, and $Z_c$ are latent tokens corresponding to context frames. The MVM training objective is different from the causal autoregressive training objective: $\mathcal{L}_{\text{AR}} = - \mathbb{E}_{x \in \mathcal{D}} \left[ \sum_{\forall i \in Z_p} \log p(z_i | z_{j<i}, Z_c) \right]$. The key difference is that for MVM the conditional dependence is *bidirectional*: *all* masked tokens are predicted conditioned on *all* tokens.

### 3.3 BIDIRECTIONAL WINDOW TRANSFORMER

Transformer models composed entirely of global self-attention modules incur significant compute and memory costs, especially for video tasks. To achieve more efficient modeling, we propose to compute self-attention in windows, based on two types of non-overlapping configurations: 1) Spatial Window (SW): attention is restricted to all the tokens within a subframe of size $1 \times h \times w$ (the first dimension is time); 2) Spatiotemporal Window (STW): attention is restricted within a 3D window of size $T \times h' \times w'$. We sequentially stack the two types of window configurations to gain both *local* and *global* interactions in a single block (Fig. 1, a, center) that we repeat $L$ times. Surprisingly, we find that a small window size of $h' = w' = 4$ is sufficient to learn a good video prediction model while significantly reducing memory requirements (Table 2b). Note that our proposed block enjoys global interaction capabilities without requiring padding or cyclic-shifting like prior works (Liu et al., 2021a;b), nor developing custom CUDA kernels for sparse attention (Child et al., 2019) as both window configurations can be instantiated via simple tensor reshaping.

### 3.4 ITERATIVE DECODING

Decoding tokens autoregressively during inference is time-consuming, as the process scales linearly with the number of tokens, and this can be prohibitively large (e.g., $4,096$ for a video with $16$ frames and $256$ tokens per frame). Our video prediction training task allows us to predict future video frames via a novel iterative non-autoregressive decoding scheme: inspired by the forward diffusion process in diffusion models (Ho et al., 2020; Nichol & Dhariwal, 2021) and the iterative decoding in generative models e.g., *mask-predict* Ghazvininejad et al. (2019) and MaskGiT Chang et al. (2022), we predict videos in $T$ steps where $T << N$, the total number of tokens to predict.

Concretely, let $\gamma(t)$, where $t \in \left\{ \frac{0}{T}, \frac{1}{T}, \ldots, \frac{T-1}{T} \right\}$, be a mask scheduling function (Fig. 3) that computes the mask ratio for tokens as a function of the decoding steps. We choose $\gamma(t)$ such that it is monotonically decreasing with respect to $t$, and it holds that $\gamma(0) \to 1$ and $\gamma(1) \to 0$ to ensure that our method converges. At $t = 0$, we start with $Z = [Z_c, Z_p]$ where all the tokens in $Z_p$ are [MASK] tokens. At each decoding iteration, we predict *all* the tokens conditioned on *all* the previously predicted tokens. For the next iteration, we mask out $n = \lceil \gamma(\frac{t}{T})N \rceil$ tokens by keeping all the previously predicted tokens and the most confident token predictions in the current decoding step. We use the softmax probability as our confidence measure.

Empirically we found that selecting the most confident tokens while performing iterative decoding generates videos with little or no motion. Intuitively, given only a few $(1-2)$ context frames, selecting the most confident tokens results in videos where the context frames are copied across all time steps. Hence, we add temperature annealed Gumbel noise to the token confidence to encourage the model to produce more diverse outputs. Concretely, let $C^t \in R^N$ be the vector of softmax probabilities of the sampled tokens $Z_p^t$ at decoding iteration $t$. We select the most confident tokens to keep at iteration $t$ from $C_g^t$ where, $C_g^t = C^t + \text{Gumbel}(\mathbf{0}, \mathbf{1}) \cdot \left(1 - \frac{t}{T}\right)$.

## 4 EXPERIMENTAL EVALUATION

In this section, we evaluate our method on three different datasets and compare its performance with prior state-of-the-art methods, using four different metrics. We also perform extensive ablation studies of different design choices, and showcase that the speed improvements due to iterative decoding enable real-time planning for robotic manipulation tasks. For qualitative results, see § B.2 and videos on our project website.

| RoboNet | param. | FVD↓ | PSNR↑ | SSIM↑ | LPIPS↓ |
|---|---|---|---|---|---|
| SVG | 298M | 123.2 | 23.9 | 87.8 | 0.060 |
| GHVAE | 599M | 95.2 | 24.7 | 89.1 | 0.036 |
| FitVid | 302M | **62.5** | **28.2** | **89.3** | **0.024** |
| MaskViT | 257M | 133.5 | 23.2 | 80.5 | 0.042 |
| MaskViT (256) | 228M | 211.7 | 20.4 | 67.1 | 0.170 |

| KITTI | param. | FVD↓ | PSNR↑ | SSIM↑ | LPIPS↓ |
|---|---|---|---|---|---|
| SVG | 298M | 1217.3 | 15.0 | 41.9 | 0.327 |
| GHVAE | 599M | 552.9 | 15.8 | 51.2 | 0.286 |
| FitVid | 302M | 884.5 | 17.1 | 49.1 | 0.217 |
| MaskViT | 181M | **401.9** | **27.2** | **58.1** | **0.089** |
| MaskViT (256) | 228M | 446.1 | 26.2 | 40.7 | 0.270 |

| BAIR | param. | FVD↓ |
|---|---|---|
| SV2P (Babaeizadeh et al., 2018) | — | 262.5 |
| LVT (Rakhimov et al., 2020) | — | 125.8 |
| SAVP (Lee et al., 2018) | — | 116.4 |
| DVD-GAN-FP (Clark et al., 2019) | — | 109.8 |
| VideoGPT (Yan et al., 2021) | — | 103.3 |
| TrIVD-GAN-FP (Luc et al., 2020) | — | 103.3 |
| VT (Weissenborn et al., 2020) | 373M | 94.0 |
| FitVid (Babaeizadeh et al., 2021) | 302M | **93.6** |
| MaskViT (ours) | 189M | **93.7** |
| MaskViT (ours, goal cond.) | 255M | 76.9 |
| MaskViT (ours, act cond.) | 255M | 70.5 |

Table 1: **Comparison with prior work.** We evaluate MaskViT on BAIR, RoboNet and KITTI datasets. Our method is competitive or outperforms prior work while being more parameter efficient.

## 4.1 Experimental Setup

**Implementation.** Our transformer model is a stack of $L$ blocks, where each block consists of two transformer layers with attention restricted to the window size of $1 \times 16 \times 16$ (spatial window) and $T \times 4 \times 4$ (spatiotemporal window), unless otherwise specified. We use learnable positional embeddings, which are the sum of space and time positional embeddings. See § A.1 for architecture details and hyperparameters.

**Metrics.** We use four evaluation metrics to compare our method with prior work: Fréchet Video Distance (FVD) (Unterthiner et al., 2018), Peak Signal-to-noise Ratio (PSNR), Structural Similarity Index Measure (SSIM) (Wang et al., 2004) and Learned Perceptual Image Patch Similarity (LPIPS) (Zhang et al., 2018). To account for the stochastic nature of video prediction, we follow prior work (Babaeizadeh et al., 2021; Villegas et al., 2019) and report the best SSIM, PSNR, and LPIPS scores over 100 trials for each video. For FVD, we use all 100 with a batch size of 256. We only conducted 1 trial per video for evaluating performance on the BAIR dataset (Ebert et al., 2017).

## 4.2 Comparison with Prior Work

**BAIR.** We first evaluate our model on the BAIR robot pushing dataset (Ebert et al., 2017), one of the most studied video modeling datasets (Fig. 7). We follow the evaluation protocol of prior works and predict 15 video frames given 1 context frame and no actions. The lack of action conditioning makes this task extremely challenging and tests the model's ability to predict plausible future robot trajectories and object interactions. MaskViT achieves similar performance to FitVid (Babaeizadeh et al., 2021) while being more parameter efficient, and it outperforms all other prior works (Table 1). Finally, to predict action-conditioned future frames, we linearly project the action vectors and add them to $Z$. As expected, action conditioning performs the best, with $25\%$ improvement in FVD.

**KITTI.** The KITTI dataset (Geiger et al., 2013) is a relatively small dataset of 57 training videos. We follow the evaluation protocol of prior work (Villegas et al., 2019) and predict 25 video frames given 5 context frames. Compared to other datasets in our evaluation, KITTI is especially challenging, as it involves dynamic backgrounds, limited training data, and long-horizon predictions. We use color jitter and random cropping data augmentation for training VQ-GAN and do not use any data augmentation for training the second stage. Across all metrics, we find that MaskViT is significantly better than prior works while using fewer parameters (Table 1). Training a transformer model with full self-attention would require prohibitively large GPU memory due to the long prediction horizon ($30 \times 16 \times 16 = 7680$). However, MaskViT can attend to all tokens because its spatiotemporal windows significantly reduce the size of the attention context. We also report video prediction results for the KITTI dataset at $256 \times 256$ resolution (Fig. 2, 8), a higher resolution that prior work was not able to obtain.

**RoboNet.** RoboNet (Dasari et al., 2019) is a large dataset of 15 million video frames of 7 different robotic arms interacting with objects and provides 5 dimensional robot action annotations. We follow the evaluation protocol of prior work (Babaeizadeh et al., 2021) and predict 10 video frames given 2 context frames and future actions. At $64 \times 64$ resolution, MaskViT is competitive but does

| blocks | embd. dim | FVD↓ |
|---|---|---|
| 6 | 768 | 96.6 |
| 6 | 1024 | **94.2** |
| 8 | 768 | 99.3 |
| 8 | 1024 | 99.5 |

| st window | FVD↓ | train mem. | train time |
|---|---|---|---|
| $16 \times 1 \times 1$ | 100.6 | 5.4 GB | 12.3 hr |
| $16 \times 4 \times 4$ | 96.6 | 7.0 GB | 12.5 hr |
| $16 \times 8 \times 8$ | **93.7** | 7.9 GB | 14.2 hr |
| $16 \times 16 \times 16$ | 96.6 | 11.6 GB | 27.9 hr |
| full self attn. | 98.2 | 16.4 GB | 40.3 hr |

| mask ratio | FVD↓ |
|---|---|
| 0.75 | 189.3 |
| 0.90 | 124.1 |
| 0.95 | 110.9 |
| 0.98 | 214.4 |
| 0.5 - 1 | **96.6** |

(a) **Model size**. Increasing embedding dim improves FVD.

(b) **Spatiotemporal window size**. Smaller window size is faster, memory efficient, and achieves lower FVD scores.

(c) **Mask ratio**. *Variable* masking ratio works best.

Table 2: **MaskViT ablation experiments** on BAIR. We compare FVD scores to ablate important design decisions with the default setting: 6 blocks, 768 embedding dimension (embd. dim), $1 \times 16 \times 16$ spatial window, $16 \times 4 \times 4$ saptiotemporal (st) window, and variable masking ratio. Default settings are marked in ⬛ blue .

not outperform prior works (Table 1). FVD of the VQ-GAN reconstructions is a lower bound for MaskViT. We found flicker artifacts in the VQ-GAN reconstructions, probably due to our use of per-frame latents, resulting in a high FVD score of 121 for the VQ-GAN reconstructions. MaskViT achieves FVD scores very close to this lower bound but performs worse than prior works due to temporally inconsistent VQ-GAN reconstructions. Finally, we also report video prediction results for the RoboNet dataset at $256 \times 256$ resolution (Fig. 2, 9).

## 4.3 FLEXIBLE CONDITIONING

Planning for long horizon tasks via Visual MPC can quickly become computationally intractable as the search complexity grows exponentially with the number of planning steps. Goal conditioned planning can be especially helpful in reducing the search space by breaking down tasks into sub-goals (Nasiriany et al., 2019; Pertsch et al., 2020). Methods like FitVid (Babaeizadeh et al., 2021) and prior two-stage methods (Yan et al., 2021), which rely on compression in both temporal and spatial domain, cannot predict goal conditioned video frames. In contrast, we can easily adapt MaskViT to predict goal-conditioned video frames by including the last frame in $Z_c$. Goal conditioning significantly ($18\%$ FVD improvement) improves performance without requiring any architectural change (Table 1). We believe our method could also be extended to perform video interpolation but leave that investigation to future work.

## 4.4 ABLATION STUDIES

We ablate MaskViT to understand the contribution of each design decision with the default settings: 6 blocks, 768 embedding dimension, $1 \times 16 \times 16$ spatial window, $16 \times 4 \times 4$ spatiotemporal window, and variable masking ratio (Table 2). See § B.1 and § B.2 for additional ablation studies and discussion.

**Model hyperparameters.** We compare the effect of the number of blocks and the embedding dimension in Table 2a. We find that having a larger embedding dimension improves the performance slightly, whereas increasing the number of blocks does not improve the performance.

**Spatiotemporal window (STW).** An important design decision of MaskViT is the size of the STW (Table 2b). We compare three different window configurations and MaskViT with full self-attention in all layers. Note that training a model with full self-attention requires using gradient checkpointing, which significantly increases the training time. In fact, the STW size of $16 \times 4 \times 4$ achieves better accuracy, while requiring $60\%$ less memory and speeds up training time by $3.3\times$.

**Masking ratio during training.** We find that a fixed masking ratio results in poor video prediction performance (Table 2c). A large masking ratio until a maximum of $95\%$ decreases FVD. Further increase in masking ratio significantly deteriorates the performance. A *variable* masking ratio performs best, as it best approximates the different masking ratios encountered during inference.

**Mask scheduling.** The choice of mask scheduling function and the number of decoding iterations during inference has a significant impact on image (Chang et al., 2022) and video generation quality (Fig. 3). We compare three types of scheduling functions: concave (cosine, square, cubic, exponen-

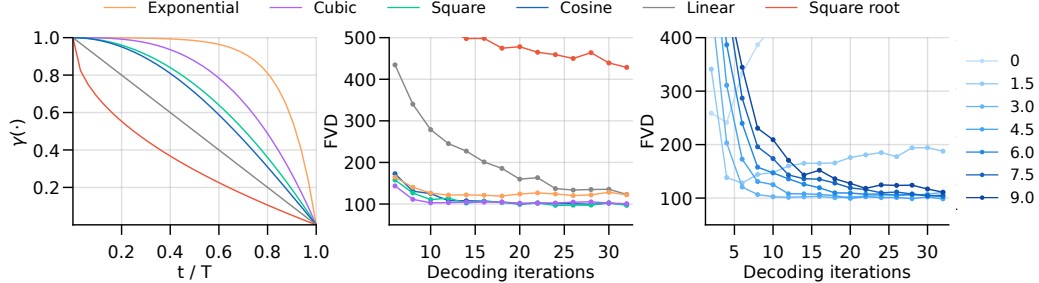

Figure 3: **Mask scheduling functions.** *Left:* 3 categories of mask scheduling functions: concave (cosine, square, cubic, exponential), linear, and convex (square root). *Middle:* FVD scores for different mask scheduling functions and decoding iterations. Concave functions perform the best. *Right:* FVD score vs. decoding iterations for different temperature values. Lower and higher temperature values lead to poor FVD scores, with a sweet spot temperature value of 3 and 4.5.

tial), linear, and convex (square root). Concave functions performed significantly better than linear and convex functions. Videos have a lot of redundant information due to temporal coherence, and consequently with only 5% of unmasked tokens (Table 2c) the entire video can be correctly predicted. The critical step is to predict these few tokens very accurately. We hypothesize that concave functions perform better as they capture this intuition by slowly predicting the initial tokens over multiple iterations and then rapidly predicting the majority of remaining tokens conditioned on the (more accurate) initial tokens in the final iterations. Convex functions operate in an opposite manner and thus perform significantly worse. Across all functions, FVD improved with increased numbers of decoding steps until a certain point. Further increasing the decoding steps did not improve FVD. Additionally, we found that selecting the most confident tokens while performing iterative decoding led to video predictions with little or no motion. Hence, we add temperature annealed Gumbel noise to the token confidence to encourage the model to produce more diverse outputs (Fig. 3). Empirically, we found that a temperature value of 4.5 works best across different datasets.

## 4.5 Visual Model Predictive Control with MaskViT on a Real Robot

In this section, we test how our generally designed framework performs when applied to real robotic tasks. We evaluate the capability of our method on the control of embodied agents through experimental evaluation on a Sawyer robot arm. We first train our model on the RoboNet dataset along with a small collection of random interaction trajectories in our setup. We then leverage MaskViT to perform visual model-predictive control, and evaluate the robot's performance on several manipulation tasks.

| dataset | pred frames | auto reg.
#  fwd. pass | ours
#  fwd. pass | speed up |
|---|---|---|---|---|
| BAIR | 15 | 3,840 | 24 | 160× |
| BAIR w/ act. | 15 | 3,840 | 12 | 320× |
| KITTI | 25 | 6,400 | 48 | 133× |
| RoboNet | 10 | 2,560 | 5 | 512× |

Table 3: **Inference speedup** of MaskViT over auto-regressive generation as measured by the number of forward passes. Iterative decoding in MaskViT can predict video frames in significantly fewer forward passes, especially when conditioned on actions.

**Setup and data collection.** We found that the VQGAN model trained only on RoboNet data did not generalize to variations in lighting, and background, or to novel objects. Therefore, we autonomously collect 120K frames of additional data with a random policy to finetune VQGAN to our setup by augmenting RoboNet. We hypothesize that this dependency on domain-specific data could be substantially reduced with a large-scale pre-training of VQ-VAEs on internet-scale data as shown by prior work (Ramesh et al., 2021), but it is beyond the scope of our experiment. At each timestep, the robot takes a 5-dimensional action representing a desired change in end-effector pose: $[x, y, z]$ gripper position, $\theta$ yaw angle, and a binary gripper open/close command. During data collection, the robot interacts with a diverse collection of random household objects (Fig. 4).

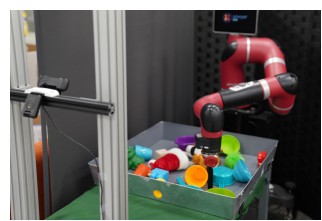 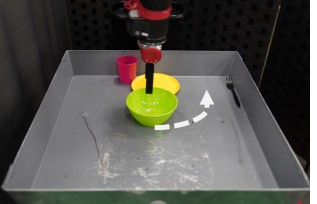

| method | success rate |
|---|---|
| MaskViT (all data) | 67% |
| FitVid (all data) | 63% |
| MaskViT (finetuned) | 53% |
| MaskViT (RN only) | 6% |
| Random policy | 3% |

Figure 4: *Left*: Third person view of real-world data collection. *Right*: An example evaluation task. The overlaid white arrow depicts the goal location of the green bowl.

Table 4: **Control evaluation results.** We perform 30 trials for each method, and report aggregated success rates.

**Model predictive control using MaskViT.** We evaluate the planning capabilities of MaskViT using visual foresight (Finn & Levine, 2017; Ebert et al., 2018) on a series of robotic pushing tasks. Our control evaluation contains two task types: `table setting`, which involves pushing a bowl to a specified location, and `sweeping`, where objects are moved into an unseen dustpan. For each task, the robot is given a $64 \times 64$ goal image and we perform planning based on MaskViT by optimizing a sequence of actions using the cross-entropy method (CEM) (Boer et al., 2004), using the $\ell_2$ pixel error between the last predicted image and the goal as the cost. We compare with two baseline methods: a FitVid model trained on combined RoboNet and the domain-specific dataset, and a random Gaussian policy. We evaluate three variants of our model: one trained on the combined RoboNet and domain-specific datasets (`all data`), one pretrained using RoboNet and then finetuned with domain-specific data (`finetuned`), and one trained only on RoboNet (`RN only`). See § A.2 for additional details and hyperparameters.

**Results.** As shown in Table 4, our model improves performance compared to FitVid on real robot experiments. Our model achieves slightly better performance when trained jointly on `all data` as opposed to pre-training on RoboNet and then finetuning on real robot data. Qualitatively, we found that domain-specific data significantly improves the scene fidelity and arm motion prediction accuracy for our model, resulting in better planning performance.

We highlight two advantages of our method when applied to real robot tasks compared to prior methods. The first advantage is the computation efficiency during inference time. Prior two-stage methods (Rakhimov et al., 2020; Yan et al., 2021) cannot be used for real-time planning due to extremely slow inference speed. Furthermore, compared to autoregressive models, MaskViT is orders of magnitude more efficient as shown in Table 3. Indeed, our method achieves $\sim 6.5$ seconds per CEM iteration. Second, compared to CNN-based architectures such as FitVid, which is the state-of-the-art method on this task, our method does not require any domain-specific inductive biases and allows for a more general inference procedure, without sacrificing any performance.

## 5 CONCLUSION

In this work, we explore MaskViT, a simple but powerful and versatile method for video prediction that leverages masked visual modeling as a pre-training task and transformers with window attention as a backbone for computation efficiency. We showed that by masking a *variable* number of tokens during training, we can achieve competitive video prediction results. Our iterative decoding scheme is significantly faster than autoregressive decoding and achieves state-of-the-art performance in planning for real robot manipulation tasks.

**Limitations and future work.** While our results are encouraging, we found that using per frame quantization can lead to flickering artifacts, especially in videos that have a static background like in RoboNet. Although MaskViT is efficient in terms of memory and parameters, scaling up video prediction, especially for scenarios that have significant camera motion (e.g., self-driving (Geiger et al., 2013) and egocentric videos (Grauman et al., 2021)) remains challenging. Finally, an important future avenue of exploration is scaling up the complexity of robotic tasks (Srivastava et al., 2022) integrating our video prediction method in more complex planning algorithms.

REPRODUCIBILITY STATEMENT

We use an open-source implementation of VQ-GAN (https://github.com/CompVis/taming-transformers) for all our experiments. We use the default parameters specified in the original implementation unless specified in Table A.1. We use PyTorch (Paszke et al., 2019) 1.7 library for implementing MaskViT. Architecture details and hyperparameters are described in § 3 and Table A.1 respectively. The inference parameters used for iterative decoding for different datasets are provided in Table A.1. Finally, please refer to the detailed description of our real robot data collection setup and experiments in § A.2.

ACKNOWLEDGMENTS

This work is in part supported by ONR MURI N00014-22-1-2740 and the Stanford Human-Centered Institute for AI (HAI).

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

## A IMPLEMENTATION DETAILS

### A.1 TRAINING MASKVIT

**VQ-GAN.** We train a VQ-GAN (Esser et al., 2021) model for each dataset which downsamples each frame into $16 \times 16$ latent codes, i.e., by a factor of $4$ for frames of size $64 \times 64$ frames and $16$ for frames of size $256 \times 256$. Table 5 summarizes our settings for all three datasets. Training VQ-GAN with discriminator loss can lead to instabilities. Hence, as suggested by (Esser et al., 2021) we start GAN losses after the reconstruction loss has converged. We also found that GAN losses were not always helpful, especially at lower input resolutions for BAIR and RoboNet.

**Transformer.** Our transformer model is a stack of $L$ blocks, where each block consists of two transformer layers with attention restricted to the window size of $1 \times 16 \times 16$ (spatial window) and $T \times 4 \times 4$ (spatiotemporal window), unless otherwise specified. We use learnable positional embeddings, which are the sum of space and time positional embeddings. Following (Liu et al., 2021a), we adopt relative position biases in our layers. We use the Adam (Kingma & Ba, 2015) optimizer with linear warmup (Goyal et al., 2017) and a cosine decay learning rate schedule. Table 5 summarizes our settings for all three datasets.

**Evaluation.** We find the optimal evaluation parameters by doing a grid search of the following parameters: $\gamma$ (cosine, square), temperature (3, 4.5) and decoding iterations depending on the prediction horizon length. We use a top-p value of $0.95$ for the BAIR dataset only. Table 5 summarizes our evaluation settings.

| Dataset | BAIR | KITTI | KITTI | RoboNet | RoboNet |
|---|---|---|---|---|---|
| Image resolution | 64 | 64 | 256 | 64 | 256 |
| Context frames | 1 | 5 | 5 | 2 | 2 |
| **VQ-GAN** | | | | | |
| Channel | 160 | 128 | 128 | 192 | 128 |
| $K$ | 1024 | 1024 | 1024 | 1024 | 1024 |
| $n_z$ | 256 | 256 | 256 | 256 | 256 |
| Batch size | 320 | 1120 | 112 | 720 | 112 |
| Training steps | 3e5 | 5e4 | 3e5 | 3e5 | 3e5 |
| Learning rate | 1e-4 | 1e-3 | 1e-4 | 5e-4 | 1e-4 |
| Disc. start | - | 2e4 | 1.5e5 | - | 1.5e5 |
| **Transformer** | | | | | |
| Spatial window | $1 \times 16 \times 16$ | $1 \times 16 \times 16$ | $1 \times 16 \times 16$ | $1 \times 16 \times 16$ | $1 \times 16 \times 16$ |
| Spatiotemporal window | $16 \times 8 \times 8$ | $16 \times 4 \times 4$ | $16 \times 4 \times 4$ | $16 \times 4 \times 4$ | $16 \times 4 \times 4$ |
| Blocks | 6 | 8 | 6 | 8 | 6 |
| Attention heads | 4 | 4 | 4 | 4 | 4 |
| Embedding dim. | 768 | 768 | 1024 | 768 | 1024 |
| Feedforward dim. | 3072 | 3072 | 4096 | 3072 | 4096 |
| Dropout | 0.0 | 0.0 | 0.0 | 0.0 | 0.0 |
| Batch size | 64 | 32 | 32 | 224 | 224 |
| Learning rate | 3e-4 | 3e-4 | 3e-4 | 3e-4 | 3e-4 |
| Training steps | 1e5 | 1e5 | 1e5 | 3e5 | 3e5 |
| **Evaluation** | | | | | |
| Mask scheduling func. | square | cosine | cosine | cosine | cosine |
| Decoding iters. | 18 | 48 | 64 | 7 | 16 |
| Temperature | 4.5 | 3.0 | 4.5 | - | - |

Table 5: **Training and evaluation hyperparameters.**

## A.2 REAL ROBOT EXPERIMENTS

**Data collection.** Our robot setup consists of a Sawyer robot arm with a Logitech C922 PRO consumer webcam for recording video frames at $640 \times 480$ resolution. All raw image observations are center-cropped to $480 \times 480$ resolution before being resized to $64 \times 64$ for model training and control in our experiments. For the finetuning dataset, we autonomously collect 5000 trajectories of 30 timesteps. At each step the robot takes a 5-dimensional action representing a change in state of the end-effector: a delta translation in Cartesian space, $[x, y, z]$, for the gripper position in meters, change in $\theta$ yaw angle of the end-effector, and a binary gripper open/close command. Following the action space used to collect the RoboNet dataset, the pitch and roll of the end-effector are kept fixed such that the gripper points with the fingers towards the table surface. Each action within a trajectory is selected independently and each dimension of the action vector is independent of the others, sampled from a diagonal Gaussian distribution, except the gripper open/close command that closes automatically when the $z$-position of the end-effector reaches below a certain threshold to increase the rate of object interaction. The random action distribution is parameterized by $\mathcal{N}(0, \text{diag}([0.035, 0.035, 0.08, \pi/18, 2])$. During data collection, we provide the robot with a diverse set of training objects to interact with. During evaluation, we test on tasks which require the robot to manipulate bowls in one setting and to push training items into an unseen dustpan in another.

**Visual-MPC.** Our control strategy is a visual MPC (Finn & Levine, 2017; Ebert et al., 2018) procedure. Given a start and goal image $I_0, I_g \in \mathbb{R}^{64 \times 64 \times 3}$, the objective is to find an optimal sequence of actions to reach the goal observation from the start. The planning objective can be written as: $\min_{a_1, a_2, ... a_H} \sum_{i=1}^{H} c_i \|\hat{f}(I_0, a_1, ...a_H)_i - I_g\|_2^2$, where $\hat{f}(I_0, a_1, ...a_H)_i$ represents the $i$th predicted frame by the learned video prediction model (MaskViT in our case), and $c_i$ are a sequence of constant hyperparameters that determine the importance of the difference between the predicted frame and the goal for each time step.

We use the cross-entropy method (CEM) (De Boer et al., 2005) to optimize a sequence of $H = 10$ future actions for this objective. In each planning iteration, we first sample $M = 256$ sequences of random actions. We then provide these sequences, together with two consecutive context frames (the previous and current step observations), and one context action (the action taken at the previous step) to MaskViT. Action sequences are sampled according to a multivariate Gaussian distribution. To bias action sampling towards smoother trajectories, the noise samples for actions in a given random trajectory are correlated across time as in Nagabandi et al. (2019). Specifically, given a correlation coefficient hyperparameter $\beta$, we first compute $u_i^1, u_i^2, ...u_i^M \overset{i.i.d}{\sim} N(0, \Sigma_i)$, where $\Sigma_i$ is the variance of the action at timestep $i$ in the current optimization iteration. The noise at timestep $i$ for the $j$th random trajectory, $n_i^j$, is then computed as a weighted combination of the new noise sample and the noise sample at the previous timestep, that is, $n_i^j = (1 - \beta) * u_i^j + \beta * n_{i-1}^j$. After all noise samples are computed, they are summed with means $\mu_i$ for each timestep, which are also iteratively updated. The final random trajectories are formed by rounding the elements in the last action dimension (gripper action) to $-1$ or $1$, whichever is closer.

Next, we compare the predictions to the goal image by computing the $\ell_2$ error and summing over time as described by the objective above. We weight the cost on the final timestep by $10\times$, but still include the costs on intermediate timesteps in the summation to encourage the robot to solve the task quickly. The best action sequences based on this score are used to refit the sampling distribution mean and variance for the next optimization iteration. After $K = 3$ optimization iterations, we execute the best scoring action sequence on the robot for the first 3 steps before performing replanning.

The robot uses a total of 15 steps to solve the task, including one initial action $= [0, 0, -0.08, 0.1, 0]$ which is executed at the beginning of every trajectory. This ensures that at least two context images provided for planning. The planning hyperparameters are summarized in Table 6.

**FitVid model training details.** For the FitVid comparison, we train FitVid on a combined dataset consisting of RoboNet and our finetuning data, just as when training our model in the `all data` case. Like in the original paper, we train the model to predict 10 future frames given 2 context frames, and use the Adam optimizer with a learning rate of $3e-4$. To be consistent with the training setup for our model, we do not use data augmentation. We train for 245K gradient steps using a batch size of 32 with 4 NVIDIA TITAN RTX GPUs.

| Hyperparameter | Value |
|---|---:|
| Total trajectory length ($T$) | 15 |
| Planning horizon | 10 |
| Number of steps between replanning | 3 |
| Action dimension | 5 |
| # of samples per CEM iteration ($M$) | 256 |
| # of CEM iterations ($K$) | 3 |
| Weights on each timestep in cost ($c_i$) | 1 if $i = 0, ...8$; 10 if $i = 9$ |
| Initial sampling distribution mean | $[0, 0, -0.5, 0, 0]$ |
| Initial sample distribution std. | $[0.05, 0.05, 0.08, \pi/18, 2]$ |
| Sampled noise correlation coefficient ($\beta$) | 0.3 |
| CEM fraction of elites | 0.05 |
| MaskViT mask scheduling function | Cosine |
| MaskViT decoding iterations | 5 |

Table 6: **Hyperparameters for visual-MPC.**

**Evaluation.** We perform control evaluation on two categories of tasks: `table setting` and `sweeping`. For each task, we test 5 different variations with 3 trials each. A trial is considered successful if the center of the object of interest is within 8 cm of the goal position after the trajectory is complete. Model inference for real robot control is performed using 8 NVIDIA RTX 3090 GPUs with a batch size of 16 per GPU. We use 5 decoding iterations, which yields a forward pass time of approximately 6.2 seconds for a batch of 256 samples.

# B  ADDITIONAL RESULTS

## B.1  QUANTITATIVE RESULTS

**Factorized attention.** An important decision decision of MaskViT is factorizing full-self attention into a block comprised of spatial and sptio-temporal window layers. This design ensures that each token can attend to *all* other tokens just as in a full-self-attention layer but without huge memory requirements. In Table 8, we compare MaskViT with a Transformer model composed entirely of STW layers. For all window sizes, the FVD scores are worse than our default block design. We note that a higher spatial resolution helps, which showcases the importance of our block design.

**VQ-GAN training dataset.** A key benefit of operating on individual frames is that we can potentially use VQ-VAEs trained on internet scale datasets (Ramesh et al., 2021). As a step towards that goal, we train a VQ-GAN model jointly on all three datasets considered in this work, i.e. BAIR, RoboNet and KITTI. During training, we create a batch consisting of equal number of frames from each dataset. We use the hyper-parameters for the RoboNet VQ-GAN model (Table 5). As shown in Table 9, across the three data sets, the performance of MaskViT trained with a shared VQ-GAN is better or similar to MaskViT trained with VQ-GAN specific to the dataset.

**Real robot experiments.** Table 7 shows per-task success rates for our real-world robotic control evaluation. Each of our two task types (table setting, sweeping) has 5 variations, each of which involves different objects to push (unseen bowls for table setting, toys previously seen in the finetuning data for sweeping) and different target locations.

## B.2  QUALITATIVE RESULTS

**Video prediction.** We present additional qualitative video prediction results for BAIR (Fig. 7), KITTI (Fig. 8) and RoboNet (Fig. 9).

**Real robot experiments.** Fig. 5 and Fig. 6 depict sample predictions for MaskViT (`all data`) and MaskViT trained only on RoboNet (`RN only`) for two example control tasks. We observe that with our model, the planner is able to find sequences of actions which bring the blue bowl or the soft red hat close to the position specified in the goal image. However, with a model which is trained only on the RoboNet dataset, planning fails. We see qualitatively that the model trained in

| Task | MaskViT (all data) | MaskViT (finetuned) | MaskViT (RN only) | Random | FitVid |
|---|---|---|---|---|---|
| *table setting (bowl color; destination)* | | | | | |
| blue; front-left | 1/3 | 1/3 | 0/3 | 0/3 | 2/3 |
| green; back-right | 1/3 | 0/3 | 0/3 | 0/3 | 0/3 |
| blue; front-right | 3/3 | 2/3 | 1/3 | 0/3 | 2/3 |
| red; front | 3/3 | 3/3 | 0/3 | 0/3 | 3/3 |
| green; left | 3/3 | 2/3 | 0/3 | 0/3 | 2/3 |
| *sweeping (object; destination)* | | | | | |
| toys; back-right | 3/3 | 3/3 | 0/3 | 0/3 | 3/3 |
| hat; front-right | 2/3 | 2/3 | 0/3 | 1/3 | 3/3 |
| hat; front-left | 1/3 | 1/3 | 1/3 | 0/3 | 2/3 |
| toys; back-left | 1/3 | 1/3 | 0/3 | 0/3 | 1/3 |
| toys; front-left | 2/3 | 1/3 | 0/3 | 0/3 | 1/3 |
| Aggregated | 20/30 | 16/30 | 2/30 | 1/30 | 19/30 |

Table 7: **Per-task quantitative results** for our robotic control evaluation. We evaluate each of 5 variants of the two tasks using 3 trials with each model or policy. Success is determined by the center of the object being within 11cm of the goal position at the end of 15 steps.

| st window | s window | FVD↓ |
|---|---|---|
| $16 \times 1 \times 1$ | - | 442.8 |
| $16 \times 4 \times 4$ | - | 242.9 |
| $16 \times 8 \times 8$ | - | 126.6 |
| $16 \times 4 \times 4$ | $1 \times 16 \times 16$ | 96.6 |

Table 8: **Block design** ablation on BAIR. We compare FVD scores for blocks consisting of only spatiotemporal window (STW) with our block design. All STW sizes perform worse due to lack of global interactions across all tokens.

| dataset | VQ-GAN | FVD↓ |
|---|---|---|
| BAIR | joint | 99.8 |
| BAIR | indv. | 96.6 |
| KITTI | joint | 388.4 |
| KITTI | indv. | 401.9 |
| RoboNet | joint | 169.6 |
| RoboNet | indv. | 133.5 |

Table 9: **VQ-GAN training dataset.** We compare using VQ-GAN models trained on individual datasets and a single VQ-GAN models trained on BAIR, KITTI and RoboNet dataset. Across all three datasets the performance of MaskViT trained with a shared VQ-GAN is better or similar to MaskViT trained with dataset specific VQ-GAN.

RoboNet-only, even when solely performing reconstruction of the first two context images using the VQ-GAN component, is unable to reconstruct the background and robot arm with high fidelity. Despite the diversity of the RoboNet dataset, finetuning on domain-specific data is still required to produce reasonable predictions in our setting.

**Discussion.** We performed extensive experiments comparing different design choices: model size, window design, masking ratio, and mask scheduling functions. Qualitatively, we found that all model sizes and different window designs work equally well, as also indicated by similar FVD scores. However, when our backbone consisted only of spatio-temporal windows, we found the performance to be much worse (Table 8). A key failure mode was lack of global consistency in the predicted video frames, e.g. the robot arm would generally disappear after the first few frames. We believe that this failure could be attributed to the lack of global interactions i.e. a token could no longer attend to all other tokens. Models trained with a low masking ratio or without adding annealed noise had a high FVD score because they showed very little or no motion of the robot arm. Finally, in comparison with other works, we found that the biggest qualitative difference was in the KITTI dataset. KITTI presents a unique challenge, as we have to predict 25 video frames. As shown in Fig. 2 and Fig. 8, MaskViT can predict future video frames without perceptually visible degradation, whereas prior work predict frames that degrade rapidly.

### B.3 LIMITATIONS AND FUTURE WORK

Although MaskViT addresses important limitations of current work in video prediction, we believe the field has not yet witnessed progress similar to text-to-image generation (Ramesh et al., 2021). Key areas of improvement are short-term temporal consistency, long-term temporal consistency, increasing the prediction horizon, and moving beyond narrow domain-specific datasets to "*in-the-wild*" settings like everyday egocentric videos (Grauman et al., 2021).

One possible way to improve short-term consistency is incorporating temporal consistency losses for VQ-GAN training. Despite using a bidirectional window attention mechanism to reduce memory requirements, we are still limited to predicting 25 frames in the future. Possible ways to predict long-horizon videos include training multiple networks at different temporal resolutions and incorporating advances in increasing the context length of Transformers. Finally, scaling both data and model parameters might be necessary to move to more realistic and complicated everyday scenes.

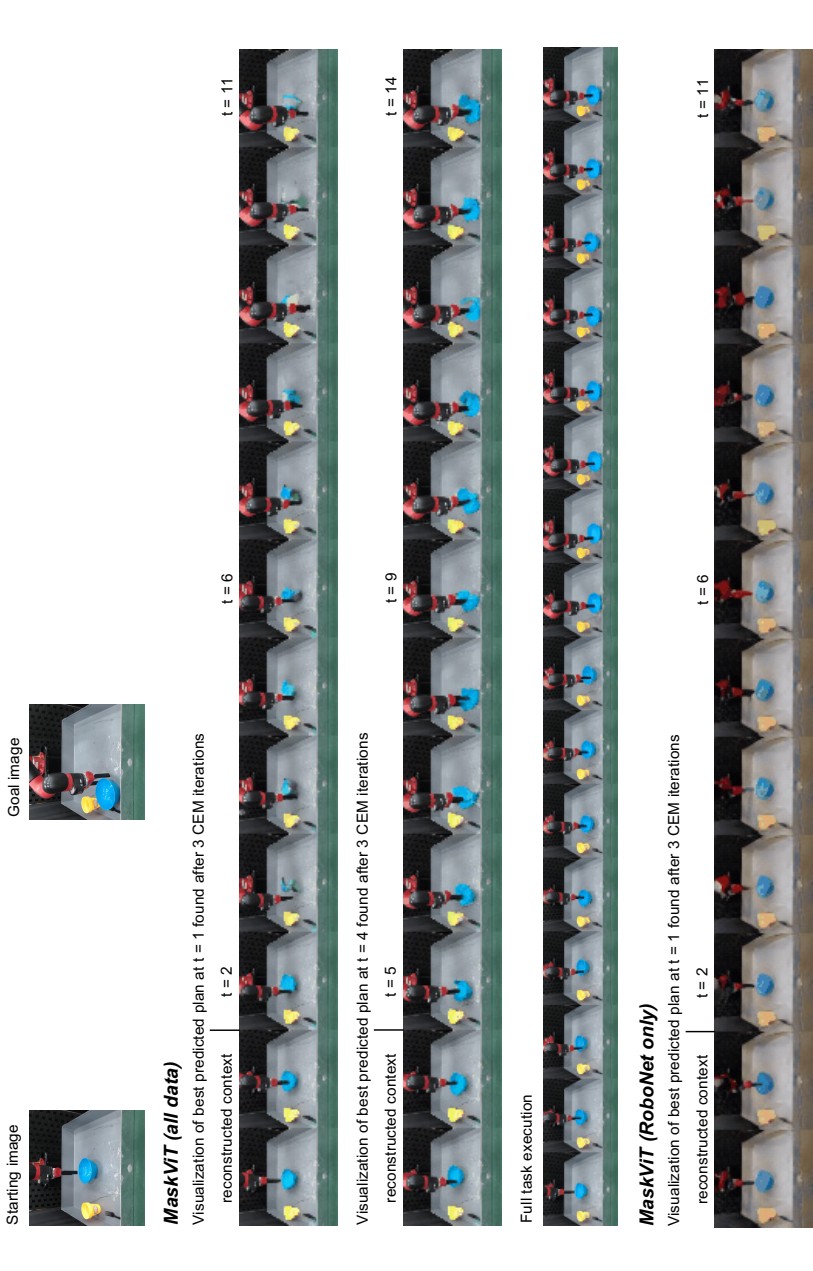

Figure 5: Visualizations for task set table: push blue bowl left.

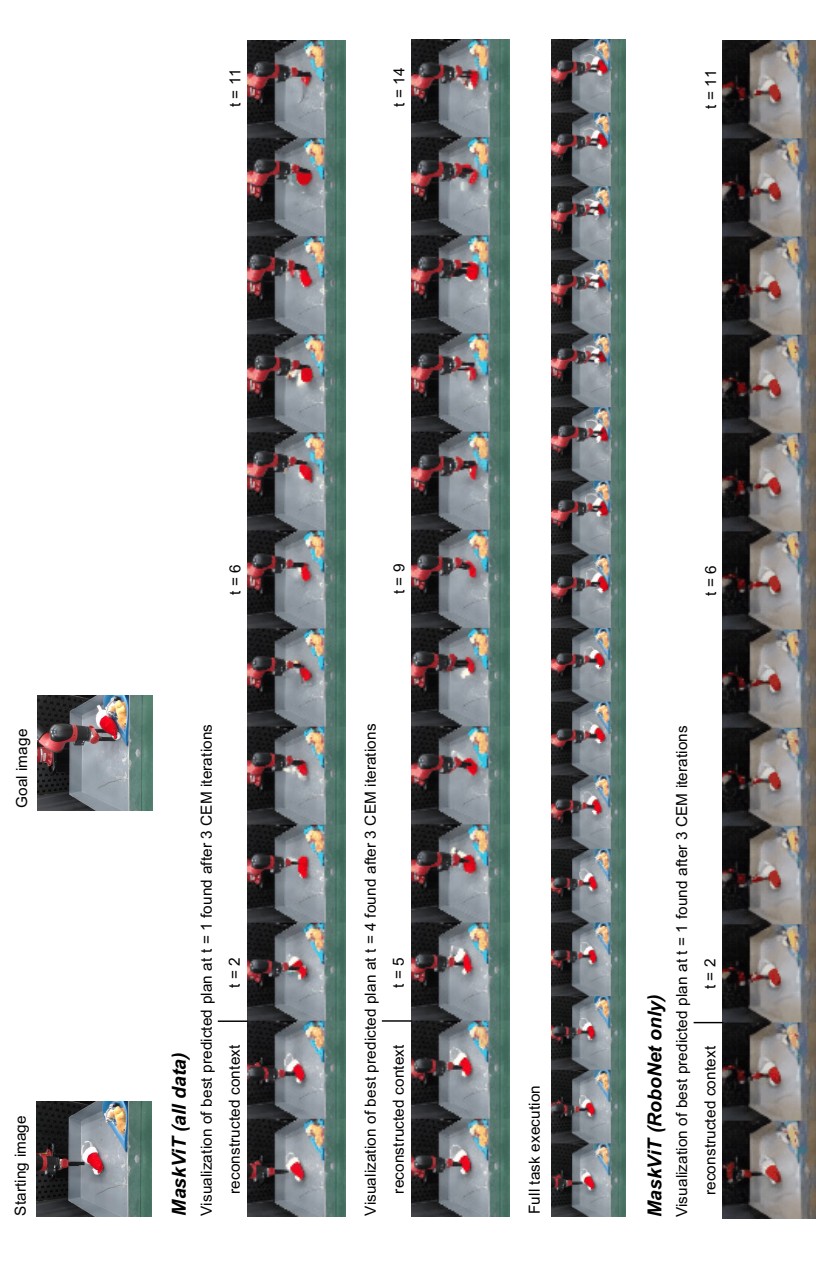

Figure 6: Visualizations for task sweep: `push hat to bottom-right corner`.

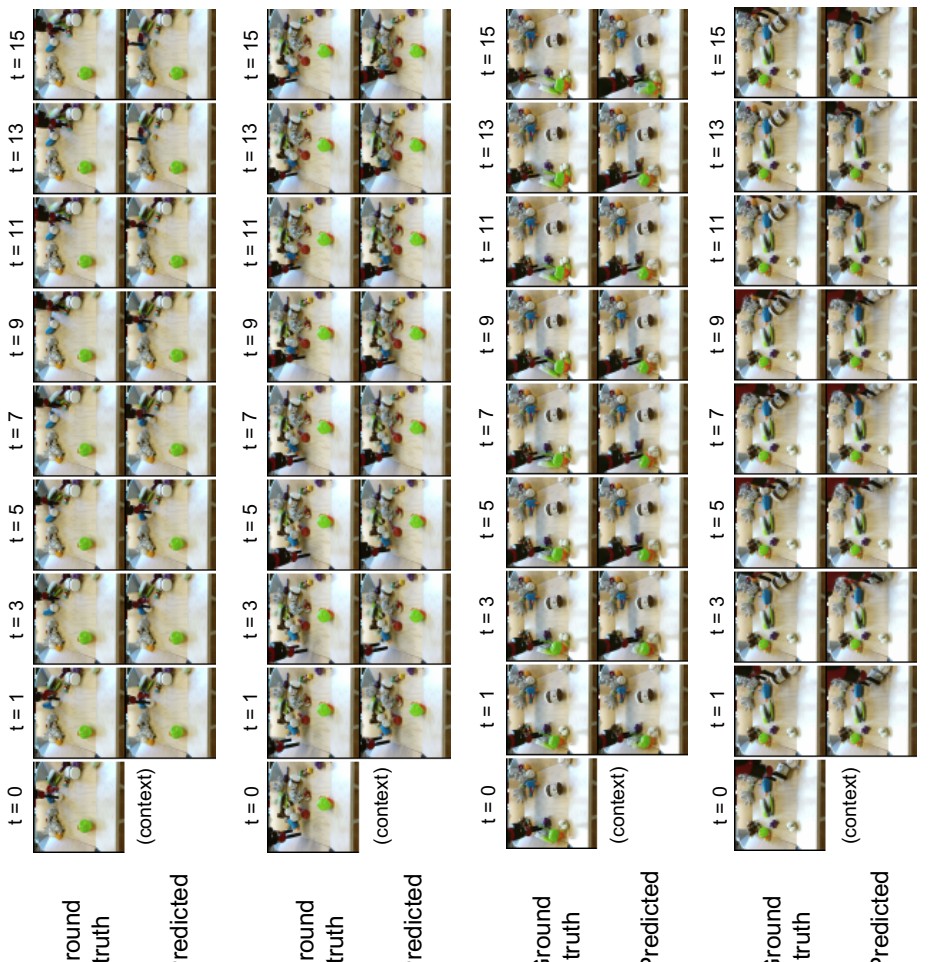

Figure 7: **Qualitative results: BAIR** (action free, $64 \times 64$)

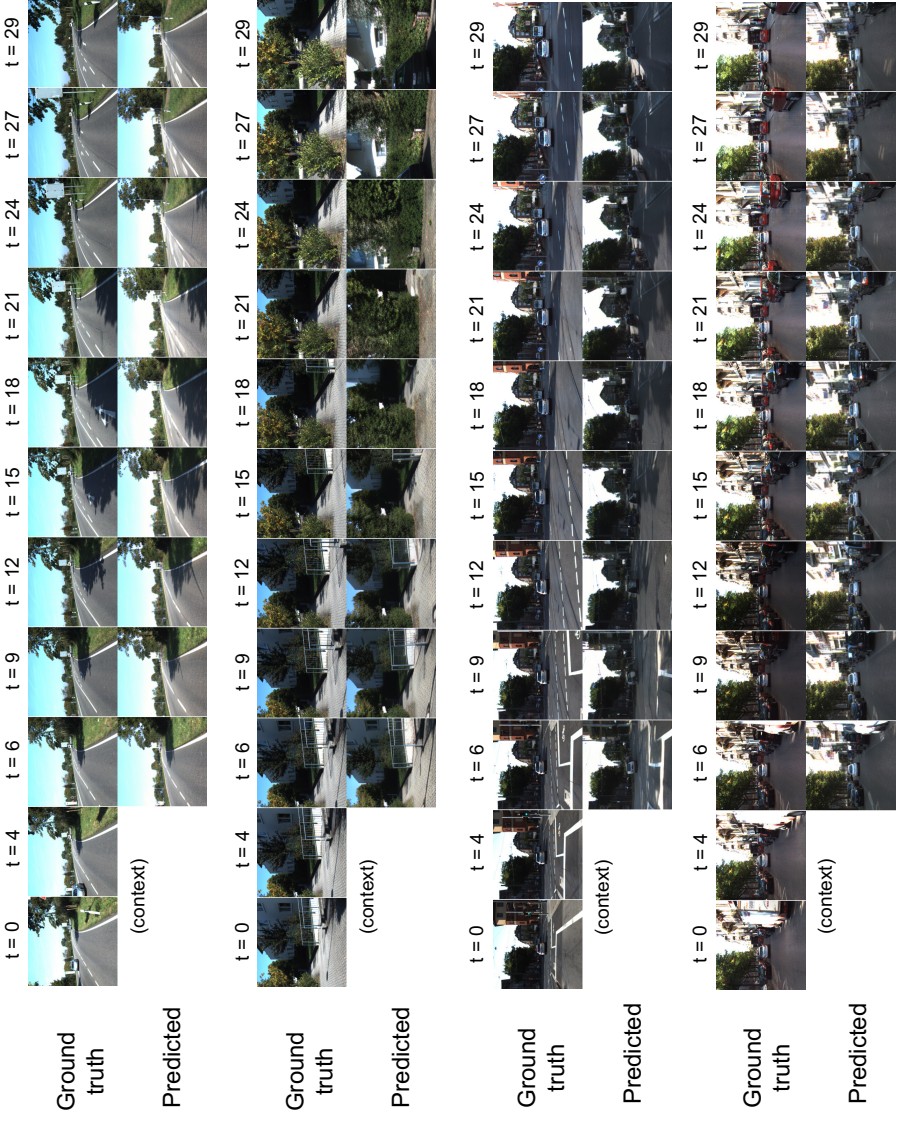

Figure 8: **Qualitative results: KITTI** ($256 \times 256$)

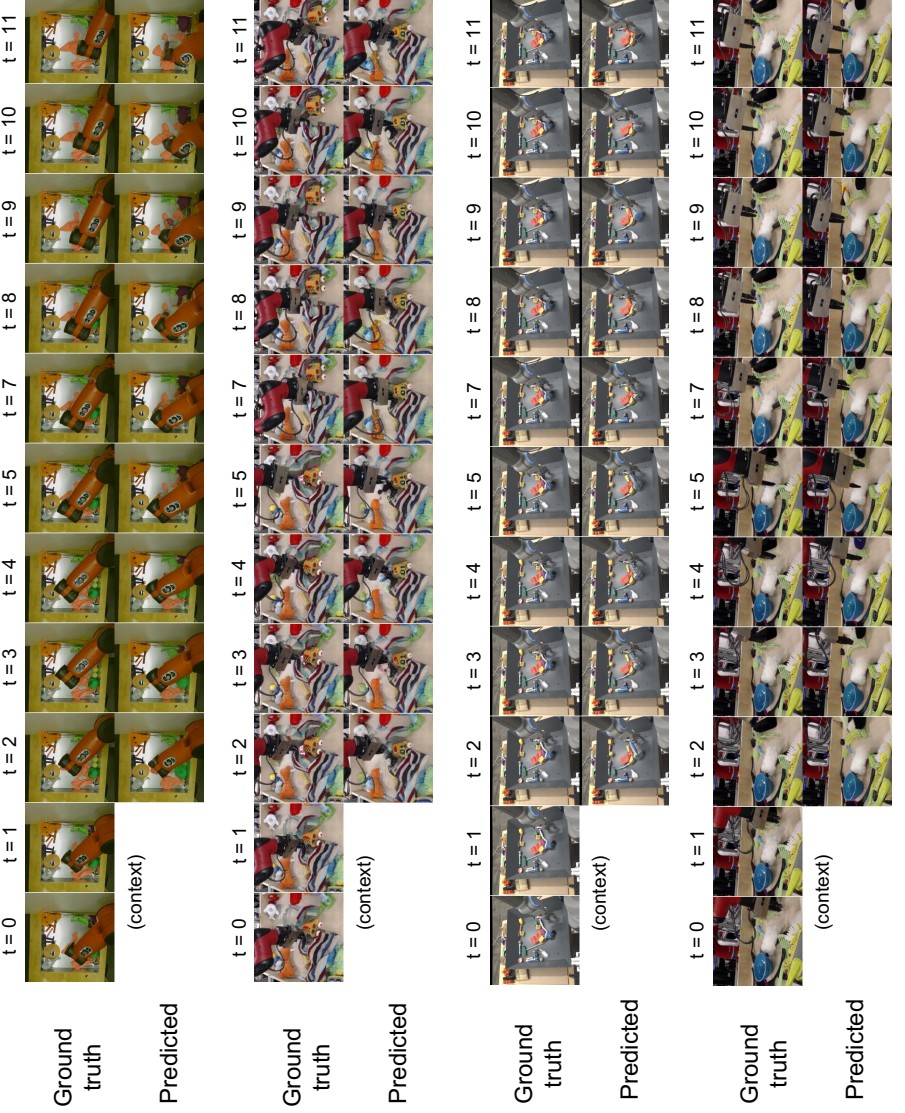

Figure 9: **Qualitative results: RoboNet** ($256 \times 256$)

