# OpenReview forum: "MaskViT: Masked Visual Pre-Training for Video Prediction"
_ICLR.cc/2023/Conference — ICLR 2023 poster_

### Official Review · Reviewer_f78f · 2022-10-24

**Confidence:** 5
**Correctness:** 3
**Technical Novelty And Significance:** 3
**Empirical Novelty And Significance:** 3
**Recommendation:** 8

**Clarity, Quality, Novelty And Reproducibility:**

## Quality
The paper has some minor faults in detail. Overall, it is well-organized and easy to follow for the readers.

## Novelty
This work is somewhat incremental to the previous works, such as masked image modeling and video prediction. However, I think this work is good because the authors combine it in a clever way and show acceptable results and valid statements in the paper.

## Reproducibility
I have no concerns about reproducibility because the proposed architecture is quite straightforward to be implemented. It would be great if the authors will release the implemented code.

## Clarity

### Related works

**Video prediction**

How about separating generative models from the video prediction part? How about describing video generation and interpolation and comparing those tasks with video prediction? It is just a proposal.

### Question on description
**line 139~140**

"The MVM training objective is different from the causal autoregressive training objective as the conditional independence is bidirectional: all masked tokens are predicted conditioned on all unmasked tokens"

To my best knowledge, all masked tokens are predicted conditioned on both all unmasked tokens and all masked tokens with embedding at each location. Am I right?

I also suggest adding a description of the causal autoregressive training objectives for comparing the MVM case.

**Strength And Weaknesses:**

## Strength
### 1. a straightforward extension of masked image modeling into the video domain

Masked image modeling (MIM) shows good performance on both representation learning via self-supervised learning and image generation. This work extends MIM into masked video modeling and effectively tackles raised problems caused by the nature of video domain.

### 2. benchmark on real robot control

Compared to the other video synthesis tasks such as video generation (no conditioned on frames), video interpolation or video temporal super-resolution, video prediction can be served as a good representation for setting like control problem or reinforcement learning. In that sense, I believe that demonstrating this kind of task is a contribution to the community of video synthesis research.

## Weaknesses
### 1. ablation study on spatial & spatial-temporal window size

The ablation study on the configuration of attention is weak.

**comparison to only spatio-temporal attention (the most important ablation)**

How about stacking single spatio-temporal attention, instead of stacking spatial and spatio-temporal attention alternatively as in the proposed architecture? I am curious about those settings because adopting homogeneous attention block is already explored in the action recognition literatures.

**more diverse setting in spatio-temporal window sizes**
Abation study on Table 2 (b) only tested the window size from (16 x 4 x 4) to (16 x 16 x 16).
So, I'm curious about the case in below,
- only temporal window (T x 1 x 1), this setting is crucial because it fully decompose temporal and spatial effect.
- different temporal window sizes while fixing the entire token numbers
   - For example, comparison with three settings (16 x 4 x 4), (4 x 8 x 8), (32 x 2 x 2) --> (last one is not available for BAIR benchmark)

### 2. a paragraph for iterative decoding policies in detail
I think that related work section (2) or iterative decoding subsection (3.4) needs a paragraph for iterative decoding policies in detail, such as MaskGIT (CVPR 22) or other relevant works. I think that comparison to relevant methods is a good way to explain the proposed one.



**Summary Of The Paper:**

This work proposes a new framework for video prediction based on masked discrete token modeling. This work extends masked image modeling (MIM) into masked video modeling with two modifications.
- First, reducing the complexity of the full attention between discrete tokens within the bidirectional transformer.
This model uses decoupled attention for spatial and temporal domains, to sparsify full attention for visual tokens.

- Second, this work introduces a masking policy during training and a decoding policy for sampling to mitigate the inconsistency between training and sampling distribution. Especially for decoding policy, this model sample causally in time direction to support generating long video sequence.

Raised concerns in the paper are verified in the experiment section. The proposed method shows comparable performance to the state-of-the-art methods. Interestingly, this work also demonstrates video prediction in a real robot control domain and achieves better performance compared to the off-the-shelf video prediction method.

**Summary Of The Review:**

The proposed methods extend masked image modeling into the video domain by tackling raised problems during extension.
The proposed method mainly design two sub-components; mixing spatial & spatio-temporal attention for bidirectional transformer and iterative decoding policy for video prediction. The choices of architecture and method design are natural and intuitive. However, I conjecture that the rationales behind the choices are not explored well in the main experiments.

---

### Official Review · Reviewer_dKZs · 2022-10-24

**Confidence:** 4
**Correctness:** 4
**Technical Novelty And Significance:** 3
**Empirical Novelty And Significance:** 3
**Recommendation:** 8

**Clarity, Quality, Novelty And Reproducibility:**

* presentation and clarity is generally very good.

* the work presents a novel patch masking, decoding scheme, and architectural components. I consider the methodological contributions as sufficiently novel.





**Strength And Weaknesses:**

**Strengths**

*S1.* Key tricks that are required to get the technique to work are presented, which is extremely useful to a reader trying to learn from this paper.

*S2.* Ablations are extensive and evaluate key design decisions.

*S3.* Appx. give appropriate details when necessary hence improving the the quality of the main paper without sacrificing clarity on key details.

*S4.* Real world robot experiments are a major strength, highlighting the applicability and usefulness of the method.

*S5.* Experiments on goal conditioning on the last frame are eliminating.

*S6.* Generality of the method to be applied to different datasets.

**Weaknesses**

*W1.* Missing some related work. I consider the masking techniques used in this paper as related to BERT-style masking that has been previously explored in the context of videos. Some relevant references to consider are Lu et al., 2019 (https://arxiv.org/abs/1908.02265), Sun et al., 2019 (https://arxiv.org/abs/1904.01766), and Zellers et al., 2021 (https://arxiv.org/abs/2106.02636). I recommend a more comprehensive review of recent related work to better contextualize the paper.

*W2.* It is not clear early in the paper why "iterative" is necessarily better than "auto-regressive." Auto-regression may also be considered an iterative process. I suggest being more precise here to give better intuition on why the proposed method is faster than existing auto-regressive methods.

*W3.* Different models (e.g., VQ-GANs) are trained for different datasets, which each represent fairly narrow domains. I would be interested to see if the technique could be extended to training a single model on a large corpus of video data or at least the union of the datasets considered. How well does this model perform compared to the single models? I feel this experiment is important to elucidate the scalability of the method.

*W4.* The intuition for why keeping the most confident frames leads to static videos is not clear to me. Given motion in the video ground truth, won't such predictions lead to very high loss?

*W5.* Why is the evaluation protocol different for BAIR than the other video datasets (L194-196)?

*W6.* Why train VQ-GAN from scratch, why not use the encoder trained on internet scale data mentioned in L291-292. Would using this tokenizer improve performance? I do not consider this to be outside the scope of the paper.

**Minor**

*M1.* I suggest changing the title as ViT usually refers to the Dosovitskiy et al., 2021 work (https://arxiv.org/abs/2010.11929), where "Vi" does not mean video. Hence the title could be a little misleading.

*M2.* The paper claims that "there is an inconsistency between the video prediction task and autoregressive masked visual pretraining – while the training process assumes partial knowledge of the ground truth future frames, at test time the model has to predict a complete sequence of future frames from scratch." However, this explanation might not be clear for someone without a lot of background knowledge on autoregressive techniques in video. I suggest giving an example or providing a reference here to make things more clear.

*M3.* I feel a major strength of the method is to be able to condition prediction on arbitrary frames in a video. This naturally allows for goal conditioning for robot manipulation, which seems like a major plus. I recommend emphasizing this feature more in the writing earlier on, otherwise it may get lost.


**Summary Of The Paper:**

The paper presents a video generation model that iteratively decodes patches in future frames. The paper also presents novel architectural components. The results are generally strong and the application of the method to a robotics task is convincing. Ablations are also strong.

**Summary Of The Review:**

I generally feel positively about the paper. The method is motivated by a need to reduce memory and improve inference time, the method seems methodologically novel, and the key contributions of the paper are empirically justified.

I currently recommend for weak acceptance of the paper.

I am most concerned about the training of different models for different datasets (see *W3*). I am also interested in the performance of using a pre-trained VQ-GAN instead of training one specifically for each video dataset (see *W6*). I feel these experiments will help provide context for the generality and scalability of the approach.

I am willing to revisit my evaluation during the rebuttal period.

POST REBUTTAL:
The authors have answered all of my concerns and added a worthwhile investigation of a shared VQ-GAN encoder. Additionally, they made it more clear that their work extends original ideas that were applied to static images. The robot experiments greatly elevate this paper. Hence, I elect to raise my score from a 6 to an 8.

---

### Official Review · Reviewer_G4sC · 2022-10-25

**Confidence:** 4
**Correctness:** 4
**Technical Novelty And Significance:** 4
**Empirical Novelty And Significance:** 4
**Recommendation:** 8

**Clarity, Quality, Novelty And Reproducibility:**

The article is generally clear and well-written, although as mentioned above, some parts would have benefited from more in-depth explanations. The description of the method is generally precise and clear but provides limited intuition in the design choices.
The quantitative analysis of the method is very thorough and convincing, but again there is limited discussion of the results from a qualitative perspective and what they mean. How far are we from having solved the problem? What are the remaining failure modes and challenges for this approach?

The proposed approach is clearly novel and progresses the field.

The description of how to reproduce the results is fairly clear, but for such a paper I would expect the code to be released nonetheless.

**Strength And Weaknesses:**

### Strengths
- This paper addresses an extremely challenging and important problem for computer vision and robotics
- The evaluation is convincing and thorough, the improvements over the state of the art follow from a clear rationale and are well discussed.
- The ablation study is well done.
- The proposed method is a clear improvement over previously published approaches.
- Applying the method to a robotic scenario, in addition to classical benchmark provides additional confidence that the method is robust.
### Weaknesses
- Although the paper is generally well written and clear, some critical aspects of the system could have been described and discussed in more details. For example 3.2 and 3.4 are essential contributions and would benefit from more explanation. Additional figures might have helped.
- The paper lacks a discussion of success and failure cases, in comparison to other approaches (and with respect to the ablation study). Because of this the paper provides little intuition on how well the method performs, beyond the (impressive) quantitative results.

**Summary Of The Paper:**

This paper tackles the problem of video prediction: given a video, the aim is to predict the future frames. This is a challenging problem as it requires learning the dynamics of the scene content in addition to the content itself. The authors argue that the classical autoregressive paradigm is both inconsistent with masked pre-training and too slow for this task.
They propose a novel approach based on: 1) a  VQGAN encoding to reduce dimensionality, 2) alternating Transformer layers based on a combination of spatial and spatiotemporal windows, 3) iterative decoding to decode latent tokens to frames, 4) variable masking during training.

The approach is evaluated on two main datasets: KITTI (driving scenes), and RoboNet (robotic manipulation), and is demonstrated to produce better predictions than the state-of-the-art with a considerably faster inference time. The approach is also used for robot predictive control.

**Summary Of The Review:**

I enjoyed reading this paper. It tackles an interesting task and describes a clear progress on this task. The paper is well written, although I would have liked more discussion of the chosen methods and of the results and their implications (possibly some quantitative results could be instead pushed in appendices), and the results are convincing.

---

### Official Review · Reviewer_V2jW · 2022-10-27

**Confidence:** 5
**Correctness:** 3
**Technical Novelty And Significance:** 2
**Empirical Novelty And Significance:** 2
**Recommendation:** 5

**Clarity, Quality, Novelty And Reproducibility:**

Clarity: The paper is very clear.

Quality: The results are clearly state-of-the-art.

Novelty: While the results are great, I think the inference speed improvement and the quality of the prediction are all coming from prior work (in particular MaskGit). I think the limited novelty is a major issue in this paper.

Reproducibility: The paper presents sufficient implementation details. I believe that reproducibility is not a concern.

**Details Of Ethics Concerns:**

I don't find ethics concerns.

**Strength And Weaknesses:**

Strength:
+ The paper is easy to read.
+ The inference speedup and training memory reduction from iterative decoding and window attention make sense. The method is technically sound.
+ The experimental results are extensive. The results demonstrate state-of-the-art performance on BAIR, RoboNet, and KITTI datasets.
+ The results show inference speedup over models with autoregressive generation (Table 3).

Weakness:
- The primary concern I have about this paper lies in its technical novelty. The core component that significantly improves the inference speed is the iterative decoding with a bidirectional transformer. This is primarily based on the work of MaskGit [Chang et al. 2022]. The MaskGit also considers multiple masking designs (Sec 3.3 in the MaskGit paper). The paper's exploration of mask scheduling (Figure 4) confirms similar findings. The variable masking ratio is also explored in MaskGit as part of the mask schedule design.

- MaskGit also considers the "conditional generation" tasks (e.g., image inpainting or extrapolation) where tokens are in the known regions. From this perspective, the proposed MaskViT is a simple adaptation of MaskGit on video data. I thus consider the novelty to be limited.

- Aside from iterative decoding, another claimed contribution is the window attention. This is interesting, but it appears to be a simple hyperparameter change with a trade-off on the training time vs. quality.

**Summary Of The Paper:**

The paper addresses the video prediction problem where the model needs to predict the future frames (or tokens) conditioned on the first frame (context). This paper specifically focuses on addressing the high memory cost and inference efficiency. The method builds upon VQGAN to encode frames into quantized tokens and learn a transformer for predicting the missing tokens. The core components of the methods are
1) spatial attention (attend only to the context frame) and spatially localized spatiotemporal attention window,
2) iterative token decoding with variable mask ratios.
Extensive results showcase improvement over the state-of-the-art on multiple datasets.

**Summary Of The Review:**

I think the method is technically sound, the results are solid, and the writing is clear. However, all these improvements are simply validating the applicability of prior work MaskGit to video data. Without careful analysis and comparison against a simple extension of MaskGit for video, it's hard to evaluate the novelty/significance of the proposed method.

---

### Decision · Program_Chairs · 2023-01-20

**Decision:**

Accept: poster

**Justification For Why Not Higher Score:**

Because of the novelty: the key idea is already proposed in a previous work.

**Justification For Why Not Lower Score:**

I think the contents of the paper is clearly and broadly beneficial to the community. So, I think it should not be rejected.

**Metareview: Summary, Strengths And Weaknesses:**

This paper proposes a masked ViT-based video prediction model. I would say that they paper is an answer to a question that how can we extend the benefits of MaskGit observed for images to videos, which does not necessarily mean that the problem is easy. Considering that the computation/space complexity of video learning is larger, it is timely and necessary study. And, this paper demonstrates the benefits well.

Strength. All reviewers agree that the paper is clear and well written. All reviewers particularly agree that it is a major strength to demonstrate on the robotics task. The claimed benefit is well demonstrated in the experiments and the rebuttal addresses the reviewer's questions well.

Weakness. The main weakness is the novelty of the main idea because one may say that it could be seen as a simple extension of MaskGit to videos. It is true and may be incremental in that sense. However, I think to study thoroughly and comprehensively a problem that the community clearly expects as a next step is also what the community benefits from. I would say that this is a good and well conducted and written paper about an unsurprising research problem.

**Note From Pc:**

if the above contains the word "oral" or "spotlight" please see: "oral" presentation means -> notable-top-5% and "spotlight" means -> notable-top-25%. As stated in our emails, we are disassociating presentation type from AC recommendations